# Leveraging transformers and explainable AI for Alzheimer's disease interpretability

**Humaira Anzum**[1], **Nabil Sadd Sammo**[1,2], **Shamim Akhter**[1]

**1** AISIP Lab, Ahsanullah University of Science and Technology, Dhaka, Bangladesh, **2** Bangladesh University of Engineering and Technology, Dhaka, Bangladesh

☯ These authors contributed equally to this work.
* nabilsadd.araf.999@gmail.com

**Data availability statement:** All relevant data are within the manuscript and its Supporting Information files.

**Funding:** The author(s) received no specific funding for this work.

## Abstract

Alzheimer's disease (AD) is a progressive brain ailment that causes memory loss, cognitive decline, and behavioral changes. It is quite concerning that one in nine adults over the age of 65 have AD. Currently there is almost no cure for AD except very few experimental treatments. However, early detection offers chances to take part in clinical trials or other investigations looking at potential new and effective Alzheimer's treatments. To detect Alzheimer's disease, brain scans such as computed tomography (CT), magnetic resonance imaging (MRI), or positron emission tomography (PET) can be performed. Many researches have been undertaken to use computer vision on MRI images, and their accuracy ranges from 80–90%, new computer vision algorithms and cutting-edge transformers have the potential to improve this performance.We utilize advanced transformers and computer vision algorithms to enhance diagnostic accuracy, achieving an impressive 99% accuracy in categorizing Alzheimer's disease stages through translating RNA text data and brain MRI images in near-real-time. We integrate the Local Interpretable Model-agnostic Explanations (LIME) explainable AI (XAI) technique to ensure the transformers' acceptance, reliability, and human interpretability. LIME helps identify crucial features in RNA sequences or specific areas in MRI images essential for diagnosing AD.

## 1 Introduction

Alzheimer's disease (AD) typically begins with mild memory loss and cognitive decline, representing a progressive and degenerative brain disorder. The absence of preventive medications utilizing modern medical technologies raises concerns about the escalating number of AD patients in the coming decades, imposing significant strain on patients, caregivers, and healthcare systems [1]. Despite the widespread prevalence and severity of the condition, effective therapy for Alzheimer's disease remains elusive. Current diagnostic tests necessitate a comprehensive understanding of the patient's medical history but often fail to provide conclusive results during the patient's lifetime [2].

Early diagnosis of AD is crucial for timely intervention. However, the diagnosis of AD is often delayed, with a substantial time gap between the onset of symptoms and confirmation. Individuals in the early stages are frequently identified with mild cognitive impairment

**Competing interests:** There are no competing interests for biasing in this work.

(MCI), a precursor to Alzheimer's disease [3]. Various assessments, including cognitive tests, blood tests, neurological evaluations [4], and brain imaging techniques such as computed tomography (CT) [5], magnetic resonance imaging (MRI) [6], and positron emission tomography (PET) [7], are employed for detection. In traditional research, manually extracting regions of interest such as the hippocampus and amygdala has been the norm for identifying AD characteristics. Recognizing AD early is also essential for enrolling patients in clinical trials or investigative studies.

Machine Learning (ML) offers a promising avenue for improving diagnostic accuracy in Alzheimer's disease. The well-established presence of neurological changes in AD-related MRI scans justifies evaluating the effectiveness of ML approaches. Many ML-based methods for AD diagnosis rely on conventional techniques like support vector machines (SVM), logistic regression (LR), linear program boosting methods (LPBM), and support vector machine-recursive feature elimination (SVM-RFE) to analyze patterns and predict AD progression. In addition, ML models can integrate gene expression data to enhance AD detection capabilities. However, traditional ML approaches often require domain-specific expertise to identify valuable features, limiting their accessibility.

Deep Learning (DL), a subset of ML, overcomes these limitations by enabling incremental feature learning. DL models are highly effective at recognizing complex patterns and processing large datasets. The traditional recurrent neural networks (RNNs) suffer from the vanishing gradient problem [8], which limits their ability to capture long-range dependencies in sequential data. A notable advancement in DL is the development of the Transformer architecture, which employs attention mechanisms to capture relationships between elements in data. Initially designed for Natural Language Processing (NLP), the Transformer's architecture has been adapted for image analysis, resulting in the Visual Transformer (ViT). ViT is better than conventional CNNs because they capture global dependencies in images through self-attention mechanisms. CNNs particularly focus on local patterns while ViTs can capture intricate patterns in an image due to their ability to their ability to extract long range dependencies across the entire image, rather than just local features like CNNs. Also, ViTs offer more flexibility and scalability, especially with large datasets. This model utilizes parallel processing to enhance speed, capacity, and accuracy in image-based applications.

Despite these advancements, the application of transformer-based models in AD diagnosis remains relatively underexplored. Notably, Convolutional Neural Networks (CNNs) and SpinalNet have demonstrated considerable accuracy in determining different stages of AD [9]. However, there is a gap in incorporating explainable artificial intelligence (XAI) techniques into transformer-based models.

This limitation highlights the need for further investigation into the integration of XAI tools to enhance interpretability and reliability in AD diagnostics. XAI can provide detailed explanations for model predictions, aiding clinicians in understanding the reasoning behind diagnostic outcomes. This is particularly important in a critical field like AD detection, where transparency and trust are essential for ensuring clinical adoption and improving patient outcomes

In this study, we aim to utilize transformer models for Alzheimer's Disease (AD) detection and evaluate their effectiveness in translating RNA text data sequences and brain MRI images. The key contributions of this work include:

(a) Implementation and evaluation of attention-based transformers with encoder-decoder architecture on AD datasets, specifically RNA sequences and MRI images.
(b) Formation of a distinctive architecture by combining vision and text transformers, designed for processing both text sequences and MRI image data.

(c) Integration of an explainable artificial intelligence (XAI) method, LIME, with the transformer. This inclusion aims to interpret the crucial features in sequence and image datasets that are essential for identifying an individual with AD.

The paper's structure is outlined as follows: Section 1 presents the introduction, Section 2 summarize previous works. In Section 3, we present the datasets, highlighting the interconnection between input attributes. Section 4 details the methodology and architecture of the proposed framework. The experimental approach, results, and analysis are elucidated in Section 5. Section 6 provides the discussion, and Section 7 concludes the study, while Section 8 briefly discusses future works in this field.

## 2 Literature review

Various earlier studies in the field have explored the prediction of Alzheimer's disease (AD) using diverse techniques, including machine learning (ML) and computer vision. In one study [10], RNA-Seq methodology demonstrated greater effectiveness than microarray analysis in assessing gene expression profiles. Machine learning models applied to Differentially Expressed Genes (DEGs) identified 740 DEGs (361 upregulated and 379 downregulated) in AD patients with varying lifespans. The Robust Rank Aggregation (RRA) technique facilitated meta-analysis of DEGs across multiple microarray platforms.

In another study [11], researchers utilized three large blood gene expression datasets (ANM1, ANM2, and ADNI) to identify genes associated with AD and classify patients using ML techniques. The study employed two procedures: extracting DEGs and feature engineering to select informative genes from the training set. Five classification techniques (logistic regression, L1-regularized logistic regression, Support Vector Machines [SVM], Random Forest [RF], and Deep Neural Networks [DNN]) were used to develop a prediction model. Both internal and external validation showed promising results, suggesting that blood-based biomarkers could advance AD diagnostics and treatments. However, challenges such as data heterogeneity, sample size variability, and RNA quality need to be addressed.

Hind Alamro et al. [12] combined three brain tissue-based AD GEO datasets, resulting in 189 AD samples and 256 non-AD samples. They identified 924 DEGs and used RF, SVM, DNN, and Convolutional Neural Networks (CNNs) for prediction. Using genes selected through LASSO and Ridge algorithms, their models achieved an Area Under the Curve (AUC) of 97.9% on independent test datasets, demonstrating the efficacy of these feature selection and classification methods.

Bhatkoti et al. [13] proposed a hybrid multi-class deep learning (DL) framework for early AD diagnosis. Their enhanced k-Sparse Autoencoder (KSA) algorithm identified degraded brain regions using MRI, cerebrospinal fluid (CSF), and positron emission tomography (PET) images from the ADNI dataset. The modified KSA achieved an accuracy of 83.143% compared to 71.327% with traditional methods. However, the need for manual tuning in determining sparsity levels (k) remains a limitation.

Tran et al. [14] introduced a computational strategy combining CNN and Gaussian Mixture Model (GMM) for brain tissue segmentation, followed by a hybrid classification model utilizing Extreme Gradient Boosting (XGBoost) and SVM. Their approach yielded high classification accuracies (0.88 and 0.80) on two datasets, with segmentation Dice coefficients of 0.96. The authors suggested further segmentation of specific brain tissues to improve precision, particularly for datasets with complex anatomical changes due to aging.

Cheng et al. [15] proposed the use of 3D-CNNs to extract features from MRI brain scans, achieving an accuracy of 87.15% and an AUC of 92.26% on the ADNI dataset. This automated method effectively identified key features for AD classification. Similarly, Isik et al. [16] achieved approximately 80% accuracy using CNNs for sMR brain images from the OASIS and MIRIAD datasets. However, they noted challenges in distinguishing mild cognitive impairment from AD.

Wang et al. [17] developed a multimodal deep learning framework for Alzheimer's disease dementia assessment, integrating data from neuroimaging, genetic markers, and cognitive tests. Their approach combines CNNs and recurrent neural networks (RNNs) to capture both spatial and temporal patterns in the data. The study shows that multimodal data fusion significantly improves diagnostic accuracy compared to single-modality approaches. The authors reported an accuracy of 92.1% for Alzheimer's disease (AD) diagnosis using their multimodal deep learning framework, which integrated neuroimaging, genetic markers, and cognitive tests. This multi-modal approach was quite a new addition to the field of AD diagnosis at the time of publication but new state-of-the-art algorithms like models can be used for further accuracy improvements.

Another study conducted by Liu M et al. [18] proposed a deep learning system for the differential diagnosis of Alzheimer's disease (AD) and mild cognitive impairment (MCI) using structural MRI. Their model leverages 3D convolutional neural networks (CNNs) to extract features from brain scans, achieving high accuracy in distinguishing between AD, MCI, and healthy controls. The study highlights the potential of deep learning in automating AD diagnosis and emphasizes the importance of structural MRI as a key biomarker for early detection. reported an accuracy of 88.6% for differentiating Alzheimer's disease (AD) from healthy controls and 76.5% for distinguishing mild cognitive impairment (MCI) from healthy controls using their deep learning system based on structural MRI. The accuracy still remains relatively low due to the use of CNN based architectures and the usability of moderately accurate model can cause serious discrepancy in the field of AD detection.

In contrast, G. Kwon et al. [19] achieved 96.12% accuracy using a CNN pipeline inspired by ResNet and ConvMixer, which reduced computational complexity while effectively classifying AD stages. Other studies, such as that by Tufail et al. [20], demonstrated the superiority of transfer learning over traditional CNNs, achieving 77.23% accuracy in binary AD classification with InceptionV3 and Xception architectures. Thamaraiselvi et al. [21] utilized DenseNet 169 for AD identification, achieving a validation accuracy of 82.23%, highlighting its efficacy in transfer learning-based approaches.

Sarwar Kamal et al. [9] explored SpinalNet and CNN for AD stage classification, achieving accuracies of 89.6% and 96.6%, respectively. They further analyzed gene expression data using SVM, k-Nearest Neighbors (KNN), and XGBoost, with SVM reaching an accuracy of 82.4%. The study employed the LIME explainable AI library to elucidate the role of genes in classification, enhancing interpretability.

Recent work by [22] applied transformer-based models, including Swin Transformer, Vision Transformer (ViT), and Bidirectional Encoder Representation from Image Transformers (BEiT), to classify Alzheimer's and Parkinson's diseases using brain imaging data. The study utilized a balanced dataset of 450 brain images, achieving classification accuracy exceeding 80%, with ViT demonstrating the highest performance (94.4% accuracy, 94.7% precision). While the results highlight the efficacy of transformer architectures in disease detection, the study has notable shortcomings. The dataset size (450 images) is relatively small, which may limit the generalizability of the findings. Also, the accuracy is not satisfactory in terms of AD disease diagnosis.

Transformer models, initially designed for natural language processing, have recently gained attention for their ability to handle long sequences through attention mechanisms and parallel processing. Despite their success in various domains, their application in AD diagnosis remains underexplored. This study addresses this gap by applying transformer-based models to both MRI and mRNA sequencing data, leveraging their capacity to identify novel biomarkers and complex patterns. Unlike conventional DL models like CNNs and Recurrent Neural Networks (RNNs), transformers overcome limitations such as vanishing gradients and sequential processing inefficiencies. Incorporating the LIME explanation technique enhances the interpretability of transformer models, bridging the gap between performance and clinical usability. An overview of the studies reviewed in this work, including their algorithms, data types, and accuracies, is summarized in Table 1, while their originality, strengths, and limitations are presented in Table 2.

## 3 Dataset description

This section provides an overview of the datasets utilized in our study, encompassing MRI images and mRNA sequencing data in tabular CSV format.

### 3.1 MRI image dataset (Alzheimer's dataset)

We utilized the Alzheimer's dataset from Kaggle [24], which includes gray MRI scans of the brain from individuals in various stages of Alzheimer's disease. The dataset encompasses four classes: MildDemented, ModerateDemented, NonDemented, and VeryMildDemented. It is organized into two folders: Train and Test, comprising a total of 5121 training images across the four classes and 1379 test images. Table ?? illustrates the distribution of images in the dataset. This dataset is valuable for predicting Alzheimer's disease stages using computer vision algorithms. Given the limited size of the test set, we allocated 10% of the training dataset for validation to ensure a robust evaluation of the model's performance. This approach was necessary because the dataset did not provide a separate validation set, and splitting the training set further would have risked overfitting due to the imbalanced class distribution. Using a small portion of the test set for validation allowed us to tune hyper-parameters and monitor model performance during training without significantly compromising the final evaluation on the remaining test data.

**Table 1. Algorithms, data type, and accuracy.**

| Author | Algorithm | Data type | Accuracies |
|---|---|---|---|
| Lee et al. [11] | L1-Regularized (LR), SVM, RF, DNN | Gene Expression | 87.40% (ANMI1), 80.40% (ANM2), 65.70% (ADNI) |
| Cheng et al. [15] | 3D-CNN | MRI Images | 87.12% (ADNI) |
| Ahsan Bil Tufail et al. [20] | 2D-CNN | MRI Images | 77.23% (OASIS) |
| Kwon et al. [19] | CNN | MRI Images | 96.12% (ADNI) |
| Isik et al. [16] | CNN | MRI Images | 80% (OASIS, MIRIAD) |
| Thamaraiselvi et al. [21] | DenseNet | MRI Images | 82.23% (OASIS) |
| Md. Sarwar Kamal et al. [9] | SpinalNet, CNN, SVC | MRI Images, Gene Expression | 89.60% (MRI), 96.60% (MRI), 82.40% (Gene) |
| Hind Alamro et al. [12] | RF, SVM, DNN, CNN | Gene Expression | 97.9% AUC |
| Bhatkoti et al. [13] | Modified KSA | MRI, CSF, PET | 83.143% |
| Tran et al. [14] | CNN, GMM, XGBoost, SVM | MRI Images | 0.88, 0.80 (Dice coefficients) |
| Wang et al. [17] | CNN, RNN | Multimodal (MRI, Genetic, Cognitive) | 92.1% |
| Liu et al. [18] | 3D-CNN | MRI Images | 88.6% (AD), 76.5% (MCI) |

**Table 2. Originality, plus aspects, and minus aspects.**

| Author | Originality | Plus Aspects | Minus Aspects |
|---|---|---|---|
| T. Lee et al. [11] | Blood-based biomarkers for AD diagnosis | High interpretability, external validation | Limited by data heterogeneity and small sample sizes |
| D. Cheng et al. [15] | Automated feature extraction from 3D MRI | High accuracy, robust feature extraction | Limited to single modality (MRI) |
| Ahsan Bil Tufail et al. [20] | Transfer learning with InceptionV3 and Xception | Reduced computational complexity | Lower accuracy compared to state-of-the-art |
| G. Kwon et al. [19] | ResNet and ConvMixer-inspired pipeline | High accuracy, reduced computational complexity | Limited interpretability of model decisions |
| Z. Isik et al. [16] | CNN for sMRI classification | Effective for sMRI data | Difficulty distinguishing MCI from AD |
| D. Thamaraiselvi et al. [21] | Transfer learning with DenseNet 169 | High efficacy in transfer learning | Moderate accuracy, limited to MRI data |
| Md. Sarwar Kamal et al. [9] | SpinalNet for AD classification, LIME for interpretability | High accuracy, explainable AI integration | Limited by dataset size and complexity |
| Hind Alamro et al. [12] | Integration of multiple GEO datasets | High AUC, robust feature selection | Limited to gene expression data |
| Bhatkoti et al. [13] | Hybrid multi-class DL framework | Improved accuracy over traditional methods | Requires manual tuning of sparsity levels |
| Tran et al. [14] | Hybrid segmentation and classification model | High Dice coefficients, effective segmentation | Complex pipeline, limited to specific datasets |
| Wang et al. [17] | Multimodal data fusion for AD diagnosis | High accuracy, leverages multiple data sources | Computationally intensive, requires full multimodal data |
| Liu et al. [18] | Deep learning for AD and MCI differentiation | High accuracy for AD, robust feature extraction | Lower accuracy for MCI, limited generalizability |

## 3.2 NCBI's RNA-sequencing datasets

In this dataset, a comprehensive collection of 191,890 nuclei associated with Alzheimer's disease (AD) has been incorporated at the single-nucleus level, providing multi-omic information. The dataset captures significant cellular heterogeneity by concurrently assessing chromatin accessibility and gene expression in the same biological samples. Leveraging single-nucleus ATAC-sequencing and RNA-sequencing, this dataset serves as a multi-omics exploration of Alzheimer's Disease in human brain tissue, accessible through the National Center for Biotechnology Information (NCBI) database under Accession Number: 174367 [25].

The assay for transpose-accessible chromatin with sequencing (ATAC-Seq) is a prevalent technique employed for evaluating chromatin accessibility throughout the genome. This method involves sequencing open chromatin regions, allowing the determination of how chromatin packaging and other factors influence gene expression. ATAC-sequencing is instrumental in detecting open chromatin regions and regulatory elements in the genome, facilitating the identification of locations potentially responsible for unregulated gene expression associated with Alzheimer's disease. The dataset integrates ATAC and RNA sequencing data, complemented by crucial factors such as age and gender, forming a binary classification dataset. This comprehensive dataset enables objective comparisons of gene expression levels

**Table 3. Dataset split for Alzheimer's disease classes.**

| Type | Moderate | Mild | Non | Very Mild |
|---|---|---|---|---|
| Train | 645 | 137 | 2304 | 1613 |
| Validation | 72 | 15 | 256 | 179 |
| Test | 179 | 112 | 640 | 448 |

between AD patients and healthy individuals, shedding light on the molecular underpinnings of the disease.

## 4 Methodology

In our methodology, we employ a unified network of vision and text transformers. To classify MRI images, we utilize a vision transformer algorithm, while a text transformer is employed for determining the presence of Alzheimer's Disease (AD) using mRNA sequence data. The vision transformer algorithm was initially introduced by Alexey Dosovitskiy et al. in [26]. These individual models are then assembled to form a cohesive transformer model, applicable for both MRI images and mRNA sequencing data. To enhance interpretability, we incorporate the LIME algorithm, providing explanations for these transformer models. This combined approach allows for a comprehensive analysis of both imaging and textual data in the context of AD diagnosis.

### 4.1 Proposed transformer model architecture

In our study, we deal with two types of input data—images and sequential data. For processing MRI images, we utilize a patch division layer that divides the images into fixed-size, non-overlapping patches, treating each patch as a meaningful unit. On the other hand, when working with mRNA sequencing data, numerical data are converted into word sequences, allowing us to represent the information as sequences of words. This dual approach, involving patch embedding for images and word sequence conversion for sequential data, enables our model to effectively handle both data types, contributing to a comprehensive analysis for Alzheimer's Disease diagnosis. Fig. 1 depitcs the unified network of Vision and text transformers.

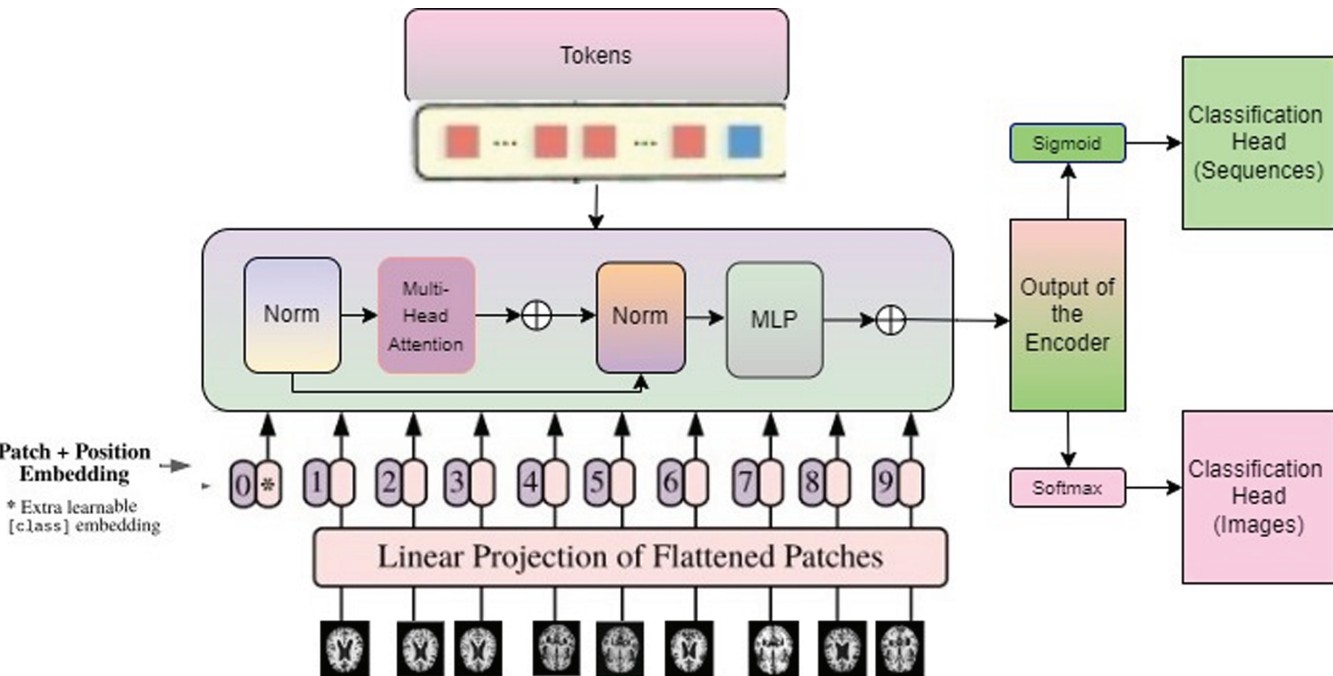

**Fig 1. The unified network of vision and text transformers.**

**4.1.1 Patch embedding (for MRI images).** In order to accommodate 2D images, we transform the image $x \in \mathbb{R}^{H \times W \times C}$ into a sequence of flattened 2D patches $x_p \in \mathbb{R}^{N \times (P^2 \cdot C)}$, where $(H = 640, W = 640)$ represents the original image resolution, $C = 3$ is the number of channels, $(P = 32, P = 32)$ is the resolution of each image patch, and $N = \frac{HW}{P^2} = 400$ is the resulting number of patches. This $N$ value also serves as the effective input sequence length for the transformer.Output class xclass is also prepared as a learnable class embedding to the sequence of embedded patches. Throughout all its layers, the transformer maintains a constant latent vector size denoted as D. Consequently, we flatten the patches and project them onto D dimensions using a trainable linear projection, as expressed by Eq. 1 [20]. Position embeddings are added to the output of this projection to retain positional information.

$$Z_0 = \left[ x_{\text{class}}; x_p^1 E; x_p^2 E; \dots ; x_p^N E \right] + E_{\text{pos}} \quad \dots \tag{1}$$

where $E \in \mathbb{R}^{(p^2 \cdot C) \times D}$, $E_{\text{pos}} \in \mathbb{R}^{(N+1) \times D}$.

**4.1.2 Token embedding (mRNA sequencing datasets).** The standard text transformer receives as input a 1D mRNA sequence of tokens, where these tokens are words or sub-words in combination with different numbers and characters. Each token is associated with an embedding vector (512 size), and an embedding layer is utilized to link each token in the input sequence to its respective embedding vector. This approach enables the model to handle continuous vector representations of discrete tokens, enhancing its ability to process and comprehend the information embedded in the input sequences more effectively. The mode_max_length attribute is set to 512, which indicates that the maximum length of the input sequence that the text transformer model can handle is 512 tokens.

**4.1.3 Positional encoding.** In the process of patch embedding, positional encoding values are computed based on the positions of patches within the image grid. Each position in the grid corresponds to a unique set of positional encoding values. For each patch embedding, these positional encoding values are element-wise added. This addition imbues the patch embeddings with information regarding their spatial positions relative to other patches in the image. Contrastingly, in the context of token embeddings, each token is assigned a distinct position. These positions signify the order of the tokens within the sequence. Positively, sinusoidal functions are employed to generate positional encodings. These functions generate a series of continuous values that smoothly change in all directions. The positional encoding values for each token embedding are then added element-wise. In both cases, the resulting sequence of embedding vectors serves as input to the encoder layer.

**4.1.4 Transformer encoder layer.** A multi-headed attention (MHA) mechanism, a 2-layer MLP, layer normalization, and residual connections are all included in the encoder component. Around each sub-layer, residual connections and layer normalization are used to improve information flow. The dimension of each sub-layer and embedding layer output is $d_{\text{model}} = 512$. We have implemented Z-Score Normalization in the hidden layer, it involves dividing each activation value by the standard deviation after subtracting the mean value from each activation. This results in a new set of activation functions with a mean of 0 and a standard deviation of 1. Notably, the normalization is conducted independently for each channel in the output tensor.

The following Eq (2) corresponds to the multi-head attention (MHA) step.

$$z_l' = \text{MHA} \left( \text{AT} \left( \text{LN} \left( z_{l-1} \right) \right) \right) + z_{l-1}, \quad l = 1 \dots L \tag{2}$$

$$\text{AT}(q, k, v) = \text{softmax} \left( \frac{q k^T}{\sqrt{d_k}} \right) v \tag{3}$$

$$z_l = \text{MLP}\left(\text{LN}\left(z_l'\right)\right) + z_l', \quad l = 1 \dots L \tag{4}$$

$$y = \text{LN}\left(z_L^0\right) \tag{5}$$

The Attention (AT) function operates within the Multi-Head Attention (MHA) function, involving three vectors: the query ($q$), key ($k$), and value ($v$). The key vector is employed to compute the attention score with the query vector, involving the transpose of the key. To normalize this computation, it is divided by the square root of $d_k$, representing the dimension of the keys. Subsequently, the softmax function is applied to the result, resulting in scaled dot product attention. These dot products are employed across six attention layers, known as heads, as detailed in Eq. 3. Eq. 4 corresponds to the step involving a multilayer perceptron (MLP). A 2-layer MLP is utilized for pre-training, and the network's final output is a vector with dimensions (1, $n_{cls}$), containing probabilities for each of the $n_{cls}$ classes.

**4.1.5 Classification head.** The classification head only uses the final representation of the patches and tokens. In the context of MRI image prediction, the softmax activation function is used due to the multi-class classification nature of the problem. For token embeddings or sequential RNA data, the activation function is sigmoid, given the binary classification nature of the problem. Eq. 5 corresponds to the output step found in each of the L-stacked transformers.

## 4.2 Model application and hyperparameter tuning

Our data is loaded from the training directory using the Data_loaders function. All images are resized to 256 × 256 pixels to ensure uniformity. Subsequently, the images were center-cropped to obtain a size of 224 × 224, as required by the vision transformer algorithm. The images are then converted to Python arrays using the to.tensor function and the array values are normalized. We utilize the timm library [27] to load a pre-trained ViT model ("vit_base_patch16_224"). To customize the model, we replace its head (topmost classification layer) with a bespoke head comprising fully connected layers. The topmost classification layer consists of a fully connected layer with some input features and 512 output features. ReLU serves as the activation function, and dropout regularization with a dropout probability of 0.3 is applied to prevent overfitting. The ReLU activation introduced non-linearity to the model.

After configuring the topmost classification head, our transformer model is created. We set a batch size of 100 for the model, using LabelSmoothingCrossEntropy as the loss function in each iteration. This loss function, tailores for classification tasks, is a modification of conventional cross-entropy and addresses overconfidence and overfitting issues during training [28]. The number of epochs is set to 30, and the Adam optimizer function is employed as the optimizer. Adam optimizes model parameters efficiently and adaptively, extending the stochastic gradient descent (SGD) method. Finally, we train our model using the model.fit method in Python. The training is conducted on a Google Colab GPU, resulting in a test average accuracy of 98.6%. Table 4 depicts the key training parameters that were used tp train the model.

## 4.3 LIME algorithm

Understanding how to articulate a predictive model and ensure precise predictions is essential. Explainable AI (XAI) is gaining traction due to its straightforward and easily understandable processes. Among the popular XAI techniques for interpreting predictive models is Local Interpretable Model-agnostic Explanations (LIME) [29]. In this context, consider X as the

**Table 4. Summary of key training parameters.**

| Parameter | Value |
|---|---|
| Image size (after preprocessing) | 224 × 224 pixels |
| Batch Size | 100 |
| Loss function | LabelSmoothingCrossEntropy |
| Number of epochs | 30 |
| Optimizer | Adam |
| Dropout probability | 0.3 |
| Activation function | ReLU |
| Pretrained model | ViT ("vit_base_patch16_224") |
| Fully connected layer output features | 512 |
| Training platform | Google Colab GPU |
| Test accuracy | 98.6% |

feature space, and x as a specific instance of a feature in the data. LIME serves the purpose of describing a prediction model. The two integral components of LIME include the black-box model (p) and the explanation (f). LIME locally elucidates the procedure by employing an interpretable function.

$$\exp(x) = \arg\min_{f \in F} \left[ \theta(p, f, \lambda_x) + \omega(f) \right] \tag{6}$$

In this context, the loss function $(p, f, \lambda_x)$ is composed of three elements: $p$, representing the black-box model; $\exp(x)$, denoting the interpretable feature explained by LIME. The term $\Omega$ accounts for the penalty associated with the complexity of the model $f$, while $\lambda_x$ represents the similarity measure between data points $x$. The function $f$ serves as the explainer. Through the utilization of anomaly data to address Eqn. 6, LIME is capable of pinpointing the features.

To elucidate the image classifications conducted by a machine learning model, we devised a Python function named show_img_exp that leverages the LIME (Local Interpretable Model-Agnostic Explanations) module. This function takes two parameters: model, representing the machine learning model to be explained, and infile, denoting the file path to the input image.

Within the function, a LimeImageExplainer object is instantiated, facilitating explanations for image classifications. The input image undergoes preprocessing through an unspecified img_prep function, and the LIME explainer is employed to generate an explanation. This explanation accentuates areas of the image primarily influencing the top predicted label in the model. To visually emphasize these significant areas, the mark_boundaries function from the scikit-image package is applied, overlaying borders on the image. This process aims to highlight regions critical to the model's top prediction. Finally, the plt.show() function is employed to display the resulting image with highlighted boundaries, enabling a clear understanding of how the model arrived at its prediction for the input image.

## 5 Results and analysis

We have obtained promising outcomes in our algorithms after completing their implementation on the datasets.

### 5.1 Results of transformer for classifying MRI images

Table 6 outlines the outcomes obtained by employing a vision transformer across various classes in the testing dataset. The NonDemented class attains the highest accuracy at 99.7%, while the MildDemented class registers the lowest accuracy at 97.31%. The vision transformer

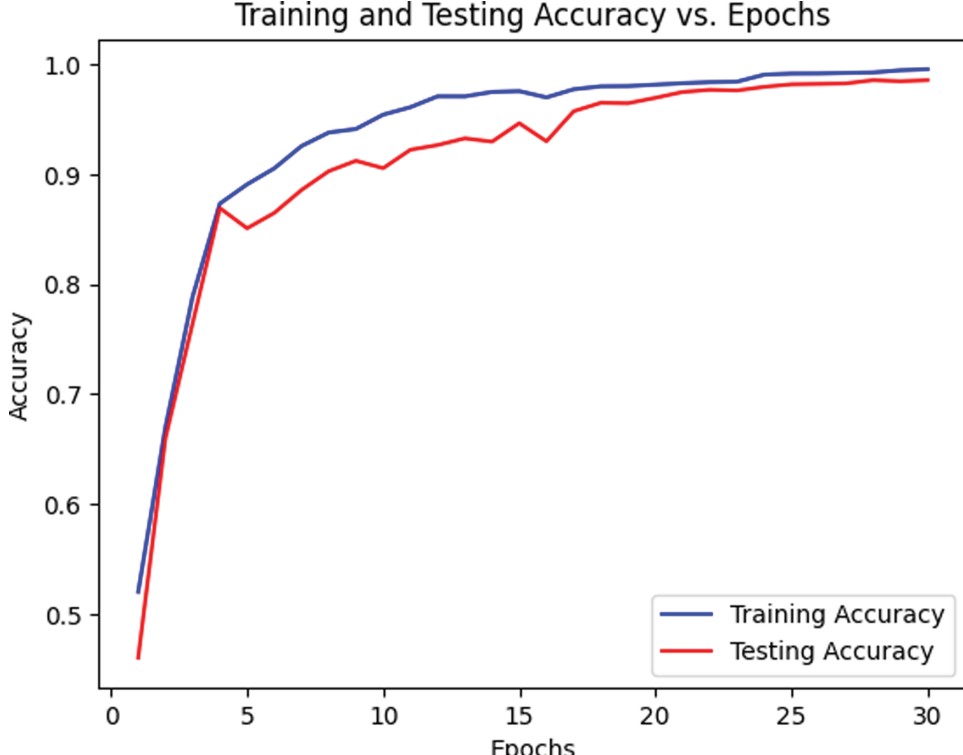

**Fig 2. Accuracy vs epoch curve of vision transformer.**

accurately classifies 98.6% of cases, with only a few instances of mis-prediction. The accuracy of the MildDemented class is less due to the presence of fewer training images in the dataset. The model performs a little less when there is a small number of training data. Other classes achieve appreciable results. Consequently, the overall accuracy of the model stands at 98.6% for all testing images, with a 95% confidence interval (CI) of [97.41%, 99.36%], indicating a high degree of reliability in classification performance. We derive the CI using 5-fold cross-validation, where the dataset was split into five subsets, training and testing the model iteratively to ensure stable performance across folds. The narrow CI range confirms robustness, as it reflects consistent accuracy with minimal variation across these folds. The CI provides a lower and upper range because it accounts for statistical uncertainty in the sample data, estimating the plausible bounds of the true accuracy. The lower bound (97.41%) represents the minimum expected performance with 95% confidence, while the upper bound (99.36%) suggests the potential peak accuracy under optimal conditions. The corresponding loss and accuracy curves are depicted in Figs 3 and 2. The accuracy curves for both training and testing exhibit an initial surge to 80% within the initial 5 epochs, eventually surpassing 95% after 15 epochs. Both curves demonstrate similar trends in loss calculations, reaching saturation after 15 epochs at the same point. Also, the ROC curve is displayed in Fig 4. The Cohen Kappa coefficient value we obtain is 0.97, indicating almost perfect agreement between the true and predicted labels. Similarly, the Matthews Correlation Coefficient (MCC) is 0.98, reflecting a strong correlation and robust performance of the model across all classes.

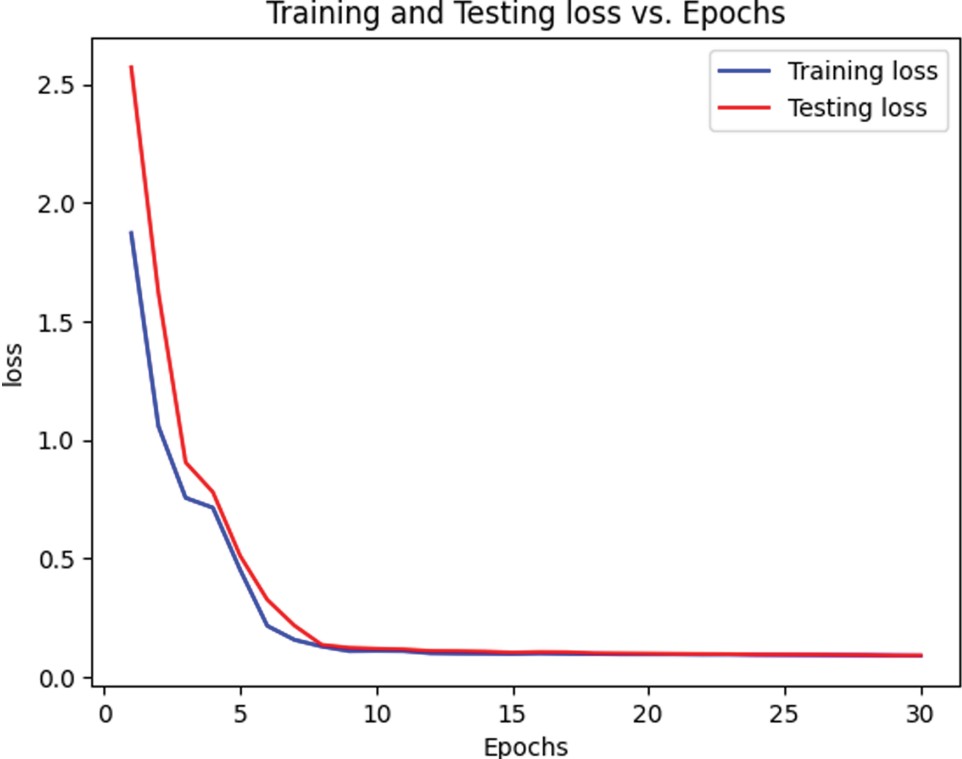

**Fig 3. Loss vs epoch curve of vision transformer.**

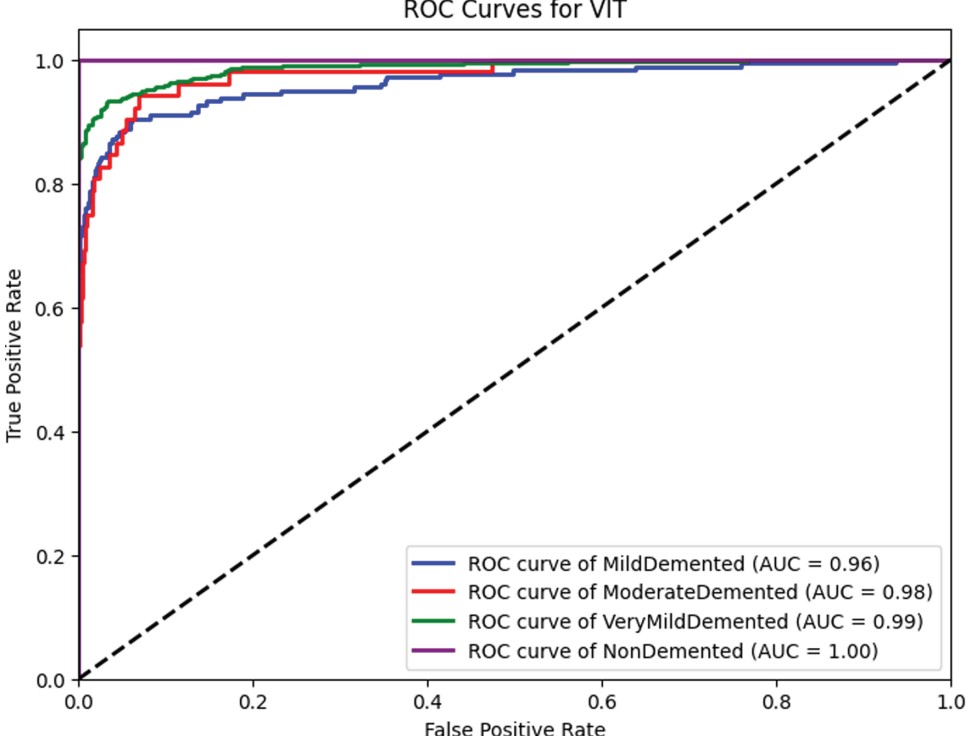

**Fig 4. AUC score of all classes of ViT.**

In Fig 5, LIME explanations are presented for all four classes of test images. The LIME explanation highlights the specific regions of the images that contributed to the prediction of these classes. Red patches in the explanation images represent areas that positively influenced the model's accurate prediction. Positive influence implies that these regions were instrumental in the model correctly identifying the class or label. In essence, the presence of red zones indicates that the characteristics or patterns in those areas align with the expected class. In Fig 6, overlays of the explanations are displayed. The yellow lines point out distinct pixels that played significant roles in predicting the class of these particular images.

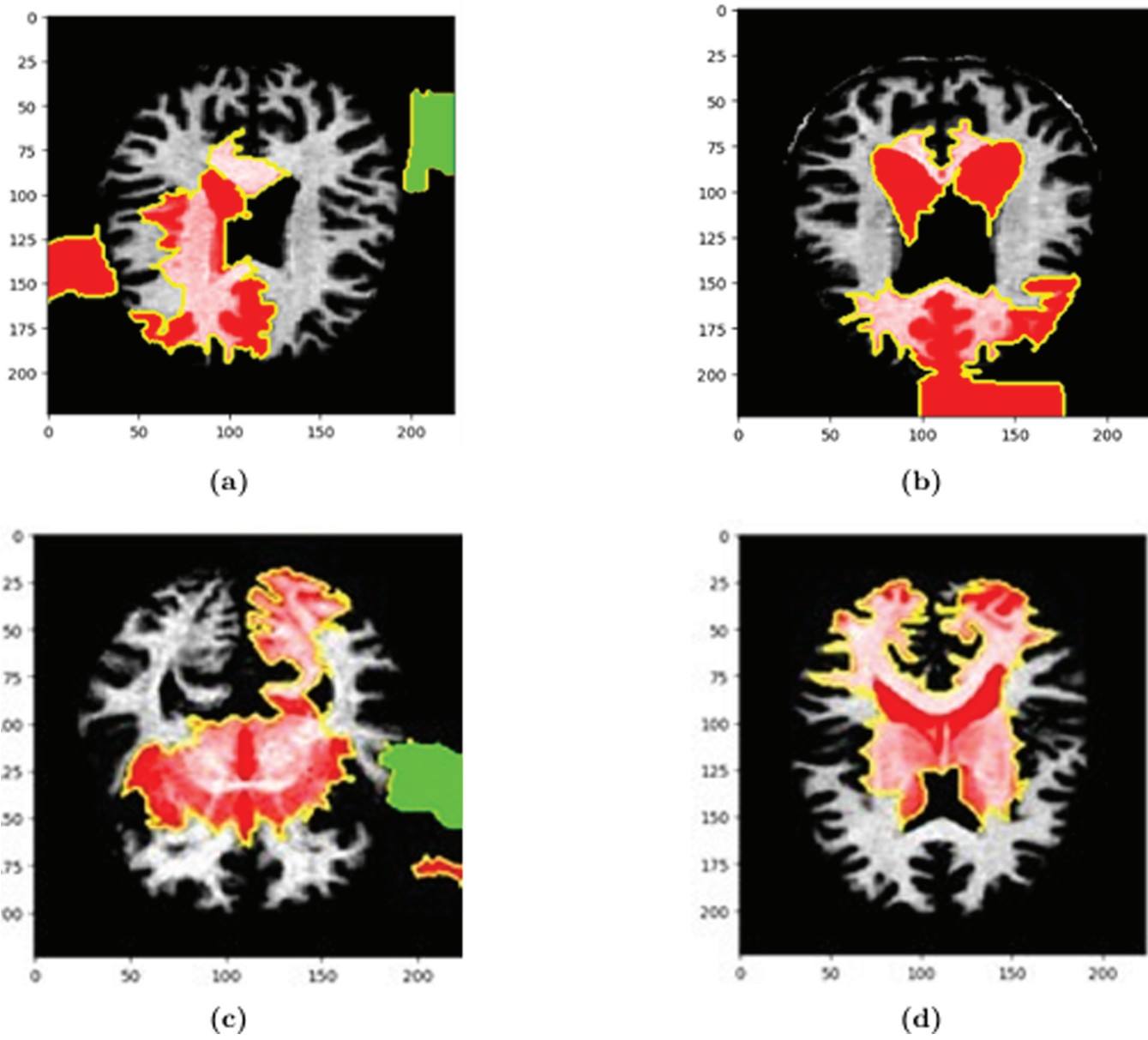

**Fig 5. LIME Explanation of (a) MildDemented, (b) ModerateDemented, (c) NonDemented, and (d) VeryMildDemented Class images.**

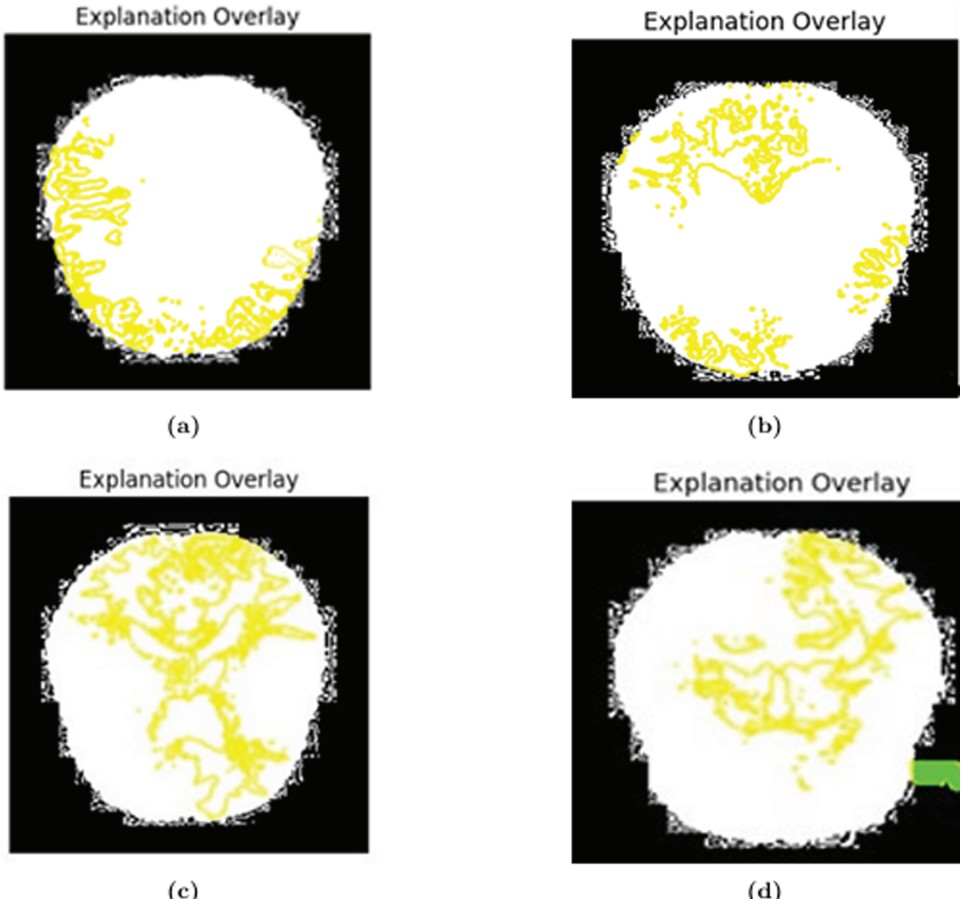

**Fig 6. Overlay of (a) MildDemented, (b) ModerateDemented, (c) NonDemented, and (d) VeryMildDemented class images.**

## 5.2 Comparative analysis of vision transformer and other state-of-the-art computer vision algorithms on MRI image classification

In our MRI image classification study, we apply four off-the-shelf computer vision algorithms which are- the vision transformer (ViT), MobileNetV2, Swin Transformer and Skip-connected Convolutional Autoencoder (SCAE). Among these algorithms, ViT and MobileNetv2, both exhibit excellent performance with only a marginal 1% accuracy difference. We select MobileNetV2 for its capability to be easily ported into mobile devices aligns well with our plans of building an embedded device for detecting Alzheimer's Disease (AD). MobileNetV2 is known for its efficiency and suitability for deployment on resource-constrained platforms, making it a practical choice for mobile applications. This decision reflects a strategic consideration for the practicality of deploying your AD detection model in real-world scenarios, especially on devices with limited computational resources. The motivation behind selecting Swin Transformer is its hierarchical feature extraction and shift-window mechanism. This mechanism particularly adept at capturing fine-grained spatial information in images, which makes it highly effective for medical imaging tasks, as reflected by its strong performance, achieving an accuracy of 91.67% on the test set.The hierarchical design of Swin Transformer makes it suitable for large-scale image classification tasks. Additionally, we apply

**Table 5. Overall test accuracy.**

| Algorithm | Accuracy | F1- score | AUC Score | Cohen's Kappa | Matthew's Coefficient |
|---|---|---|---|---|---|
| ViT | 98.60% | 98.47% | 0.992 | 0.97 | 0.98 |
| MobileNetV2 | 97.61% | 97.52% | 0.969 | 0.95 | 0.96 |
| Swin Transformer | 91.67% | 91.93% | 0.907 | 0.84 | 0.85 |
| SCAE | 88.31% | 88.83% | 0.893 | 0.78 | 0.80 |
| EfficientNet-B0 | 91.57% | 92.01% | 0.92 | 0.83 | 0.84 |
| ResNet-50 | 86.59% | 86.87% | 0.878 | 0.77 | 0.79 |

**Table 6. Test accuracy for the classes.**

| Classes | Test Accuracy | AUC |
|---|---|---|
| ModerateDemented | 99.20% | 0.98 |
| MildDemented | 97.31% | 0.96 |
| NonDemented | 99.70% | 1.00 |
| VeryMildDemented | 98.19% | 0.99 |

SCAE for its lightweight architecture and computational efficiency which gives us an accuracy of 88.31%. Additionally, EfficientNet-B0 and ResNet-50 are included for benchmarking, achieving accuracies of 91.57% and 86.59%, respectively, further validating the robustness of our approach. These results collectively highlight the trade-offs between accuracy, computational efficiency, and deployment feasibility. Based on our findings, it can be conclusively stated that the ViT proves to be a highly effective and efficient algorithm for the classification of MRI images related to Alzheimer's Disease. Table 5 displays the Accuracy, F1-score, Cohen's Kappa and Matthew's correlation coefficient of each model on the dataset. Based on our findings, it can be conclusively stated that the Vision Transformer proves to be a highly effective and efficient algorithm for the classification of MRI images related to Alzheimer's Disease. Also, Fig 4 represents the ROC curve of ViT, where we can see that ViT achieves high Area Under the Curve (AUC) scores across all Alzheimer's Disease classes, reflecting its strong ability to distinguish between positive and negative cases for each category. The AUC values 0.98 for ModerateDemented, 0.96 for MildDemented, 1.00 for NonDemented, and 0.99 for VeryMildDemented indicate near-perfect discrimination, with NonDemented reaching an ideal 1.00, aligning with its top accuracy of 99.70%. These results underscore ViT's robustness in handling class-specific patterns in brain imaging data, even for challenging cases like MildDemented as the number of training images are very low, where accuracy dips slightly and so does the AUC. Also, the Cohen's Kappa and Matthew's correlation coefficients are consistent with these findings.

## 5.3 Results of transformer for classifying RNA-sequencing

The text transformer demonstrates a 98.9% accuracy on the test RNA-Sequencing Datasets, with an 80% training and 20% testing split of the entire RNA datasets. Following this, model explanations are provided using LIME. The LIME explanation reveals the feature importance values contributing to the prediction of the probability of Alzheimer's Disease (AD). Notably, the Age attribute emerges as a highly significant factor in AD presence, while Sex shows no discernible impact. RIN, cluster, Tangle.Stage, Cell.Type, and PMI exhibit varying degrees of significance in influencing AD prediction. Additionally, a Deep Neural Network (DNN) was employed, achieving a 93.23% accuracy. Figs 7 and 8 present LIME explanations for further insights.

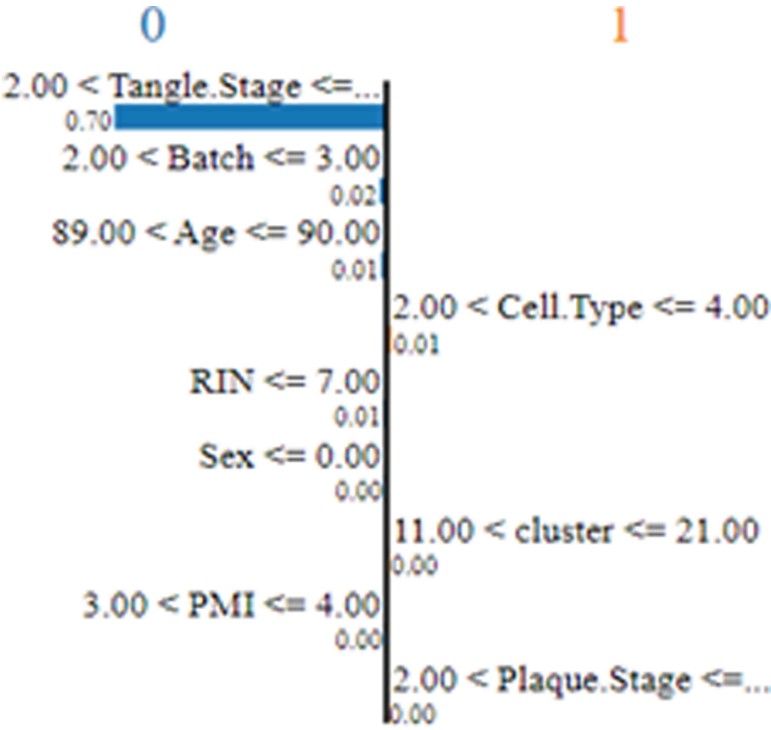

**Fig 7. LIME explanation of feature value range on prediction.**

## 5.4 Combined results of transformer models

Impressively, by employing ensemble vision and text transformers for Alzheimer's Disease (AD) prediction using both MRI images and categorical data, we attain a remarkable accuracy of 98.75%. The model demonstrates high reliability, especially in terms of accuracy. In the next two subsections, the robustness of our model is discussed.

## 5.5 Interpretation of the model from a medical perspective

From Fig 5 we can see that the regions highlighted by LIME are responsible for the respective predictions. In the image (b) the brain is moderately demented. It can be seen that the LIME explanation highlighting the ventricular and cerebellar regions in the MRI scan aligns well with known clinical biomarkers of Alzheimer's Disease (AD). Ventricular enlargement, as observed in the highlighted regions, is a well-established indicator of gray matter atrophy, particularly in the hippocampus and temporal lobes, which are critical for memory processing [30]. This degeneration seen in AD leads to cognitive impairment, and this shrinkage causes the lateral ventricles to expand [31]. The highlighted cerebellar region is also noteworthy, as recent studies suggest that cerebellar atrophy correlates with late-stage AD and cognitive decline [32]. Furthermore, cortical atrophy in the parietal and temporal lobes, also contributing to dementia symptoms, is consistent with the LIME-explained model's focus. The model's emphasis on these regions suggests that it has effectively learned clinically relevant patterns for AD detection. Also, in image (a), which corresponds to a mildly demented patient, the LIME overlay highlights critical brain regions associated with early-stage Alzheimer's Disease (AD). The hippocampus and temporal lobe regions are highlighted in red show significant importance for the model's classification. Clinically, hippocampal atrophy is a well-known

| Feature | Value |
|---|---|
| Tangle.Stage | 3.00 |
| Batch | 3.00 |
| Age | 90.00 |
| Cell.Type | 4.00 |
| RIN | 6.00 |
| Sex | 0.00 |
| cluster | 16.00 |
| PMI | 4.00 |
| Plaque.Stage | 3.00 |

**Fig 8. LIME feature impacts on prediction.**

early biomarker for AD, leading to short-term memory loss and cognitive decline [30]. Additionally, ventricular enlargement is apparent, likely due to gray matter loss. As a result, cerebrospinal fluid (CSF)-filled spaces will expand [31]. In image (c), corresponding to a non-demented individual, the LIME explanation shows activation in white matter regions and cerebellum. The absence of significant highlights in the hippocampus or temporal lobe suggests a healthy brain structure with no signs of neurodegeneration. The mild highlighting in the cerebellum and cortical areas could be due to normal variations in MRI signal intensity or non-pathological age-related changes [32]. Unlike the demented case, the ventricular spaces appear normal, indicating that the brain volume is well-preserved. Thus it can be concluded based on clinical research evidence that, the LIME explanation supports the validity of the model. For the RNA sequencing data, Fig 7 illustrates a LIME-based decision boundary, showing how different features contribute to predicting Alzheimer's Disease (AD). The Tangle.Stage, Plaque.Stage, and Cell.Type features play a significant role, as neurofibrillary tangles and amyloid plaques are established biomarkers of AD pathology [31]. Fig 8 highlights the feature importance ranking, where Tangle.Stage (3.00) and Plaque.Stage (3.00) have strong contributions, supporting their role in AD progression. Age (90 years) and cluster (16.00) also impact prediction, aligning with research showing that age-related neuronal changes increase AD risk [30]. The model's reliance on these biologically relevant features suggests clinically meaningful predictions, reinforcing the validity of LIME explanations.

## 5.6 Additional proof of robustness

**5.6.1 Validation using ADNI dataset** After training our model using the ADNI dataset, we receive remarkable accuracy of 98.29% in the test dataset with the same hyperparameters and configuration. The accuracies of different classes are depicted in the Table 7.

**5.6.2 LIME interpretation** The model's robustness is reinforced through LIME-based interpretability, which highlights clinically significant brain regions associated with Alzheimer's Disease (AD). The ventricular enlargement, hippocampal atrophy, and cerebellar atrophy identified in the LIME explanations align with well-documented biomarkers of AD. These findings confirm that the model is not relying on spurious correlations but instead learning meaningful anatomical features relevant to AD progression. Additionally, the LIME interpretations for non-demented cases show no significant highlighting in the hippocampus or temporal lobe, further validating the model's ability to differentiate between pathological and healthy brain structures. Furthermore, in the RNA sequencing data, the model correctly emphasizes Tangle.Stage and Plaque.Stage, which are well-known indicators of AD pathology. The alignment between model predictions and established clinical knowledge supports the model's reliability and generalizability, demonstrating its robustness in detecting AD across different modalities.

## 6 Discussion

The findings of this study underscore the transformative potential of advanced deep learning models, such as Vision Transformer (ViT) and MobileNetV2, in enhancing the accuracy of Alzheimer's disease (AD) diagnosis through MRI image analysis. ViT's superior performance, achieving an overall accuracy of 98.6% with a tight 95% confidence interval [97.41%, 99.36%], highlights its ability to capture intricate patterns in brain imaging data that traditional methods might overlook. MobileNetV2, while slightly less accurate, offers a lightweight alternative, making it viable for resource-constrained settings, which broadens the practical scope of AI-driven diagnostics. The integration of Explainable AI (XAI) techniques like LIME further elevates these models by providing interpretable insights, such as highlighting the hippocampus and temporal cortex as key regions in AD progression. This interpretability bridges the gap between complex AI outputs and clinical utility, fostering trust among healthcare professionals who rely on actionable explanations. Extending the analysis to RNA-sequencing data with transformer learning revealed complementary insights, suggesting that multi-modal approaches could refine AD prediction beyond imaging alone. However, the study's reliance on specific datasets raises concerns about generalizability, as demographic imbalances—age, gender, or ethnicity—may skew model performance across diverse populations. Similarly, variations in MRI equipment and protocols across institutions introduce noise, potentially undermining the models' consistency in real-world scenarios. These limitations echo broader challenges in AI healthcare research, where dataset quality often dictates success more than algorithmic sophistication. The high AUC scores across AD stages

**Table 7. Class-wise accuracy of ADNI dataset.**

| Class | Accuracy (%) |
|---|---|
| CN (Cognitively Normal) | 99.1 |
| EMCI (Early Mild Cognitive Impairment) | 97.3 |
| LMCI (Late Mild Cognitive Impairment) | 98.5 |
| AD (Alzheimer's Disease) | 98.29 |

(e.g., 1.00 for NonDemented, 0.96 for MildDemented) from ViT suggest robust discrimination, yet they also prompt questions about overfitting to the training data's characteristics. From a clinical perspective, aligning AI outputs with known AD biomarkers, like hippocampal atrophy, strengthens diagnostic confidence, but validation against larger, standardized cohorts remains critical. The regulatory landscape adds another layer of complexity, as FDA approval demands rigorous, multi-center trials to ensure safety and efficacy beyond academic benchmarks. Encouragingly, the narrow confidence intervals from 5-fold cross-validation affirm the models' stability, yet future work must address scalability to diverse clinical environments. Integrating multi-center datasets with standardized protocols could mitigate biases and enhance deployment feasibility. Ultimately, this study lays a foundation for AI to revolutionize AD diagnostics, but its success hinges on overcoming technical, ethical, and regulatory hurdles in tandem.

## 7 Conclusion

This study demonstrates the efficacy of employing ensemble vision and text transformer model enhanced by XAI techniques like LIME, in accurately diagnosing AD stages using both MRI and RNA-sequencing data. The high accuracy (98.75%) and interpretability underscore their potential to improve clinical understanding and early detection of AD. Our experiments also highlight the robustness, absence of overfitting and tendency to bias towards different types of data. However, noisy dataset and variations of MRI protocol may highlight our model's limitations that future work must address. Regulatory challenges, including FDA approval, remain significant for real-world adoption. These findings pave the way for advanced AI tools in AD management, pending broader validation and standardization.

## 8 Future works

In the future, the progression of this research could involve the development of an embedded hardware system employing Vision Transformer techniques to enhance the accuracy of MRI image predictions. Additionally, the inclusion of diverse and standardized datasets for model training could lead to the creation of a more generalized and efficient system, improving overall prediction accuracy. Furthermore, utilizing the MobileNetV2 algorithm, there is potential for the creation of a mobile cloud application dedicated to Alzheimer's Disease detection using MRI images.

## Author contributions

**Conceptualization:** Humaira Anzum, Nabil Sadd Sammo.

**Formal analysis:** Nabil Sadd Sammo.

**Investigation:** Nabil Sadd Sammo.

**Methodology:** Humaira Anzum, Nabil Sadd Sammo.

**Project administration:** Nabil Sadd Sammo.

**Software:** Nabil Sadd Sammo.

**Supervision:** Shamim Akhter.

**Validation:** Humaira Anzum, Nabil Sadd Sammo, Shamim Akhter.

**Visualization:** Humaira Anzum.

**Writing – original draft:** Humaira Anzum, Nabil Sadd Sammo.

**Writing – review & editing:** Nabil Sadd Sammo, Shamim Akhter.

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
