## [Decision Letter · Decision Letter 0]

25 Nov 2024

PONE-D-24-39865Leveraging Transformers and Explainable AI for Alzheimer’s Disease InterpretabilityPLOS ONE

Dear Dr. Sammo,

Thank you for submitting your manuscript to PLOS ONE. After careful consideration, we feel that it has merit but does not fully meet PLOS ONE’s publication criteria as it currently stands. Therefore, we invite you to submit a revised version of the manuscript that addresses the points raised during the review process.

We look forward to receiving your revised manuscript.

Kind regards,

Laith Alzubaidi

Academic Editor

PLOS ONE

Journal Requirements:

2. Please note that PLOS ONE has spec6ific guidelines on code sharing for submissions in which author-generated code underpins the findings in the manuscript. In these cases, all author-generated code must be made available without restrictions upon publication of the work. Please review our guidelines at https://journals.plos.org/plosone/s/materials-and-software-sharing#loc-sharing-code and ensure that your code is shared in a way that follows best practice and facilitates reproducibility and reuse.

5. Please ensure that you refer to Figure 1 in your text as, if accepted, production will need this reference to link the reader to the figure.

Reviewers' comments:

Reviewer's Responses to Questions

**Comments to the Author**

1. Is the manuscript technically sound, and do the data support the conclusions?

Reviewer #1: Partly

Reviewer #2: Yes

2. Has the statistical analysis been performed appropriately and rigorously? 

Reviewer #1: Yes

Reviewer #2: Yes

3. Have the authors made all data underlying the findings in their manuscript fully available?

Reviewer #1: Yes

Reviewer #2: Yes

4. Is the manuscript presented in an intelligible fashion and written in standard English?

Reviewer #1: No

Reviewer #2: Yes

5. Review Comments to the Author

Reviewer #1: The authors of this manuscript present an interesting approach for the interpretability of Alzheimer’s Disease. While the methodology and concept are effective, I have a few suggestions on various sections of the paper that may require improvements to meet the journal's standards.

1. First, I recommend that the authors consider structuring the introduction into several segments based on relevance. While the introduction begins well, the connections between sentences become weaker as it progresses. Additionally, there are insufficient citations for various claims presented in this section. The overall writing quality of the introduction could be improved. For example, the statement, 'However, there is a gap in incorporating XAI techniques into transformer-based models,' is made but is not followed by an explanation or discussion of the gaps, nor is there any citation provided.

2. Similar improvements are needed in the literature review. While the content is generally acceptable, the writing style should be enhanced. It would be beneficial to ensure relevance between sentences and to review the word choices throughout. For instance, if you spell out 'Convolutional Neural Network' once, you can simply use the abbreviation 'CNN' in subsequent references.

3. The results section requires additional validation considering the claims made by the authors. Currently, the authors only performed a comparative analysis with DeepLabV2. I suggest including a few more off-the-shelf network architectures to better validate your results.

4. Quality of Figure 6 and 7 can be improved.

5. Check all the section heading for inaccurate writing format.

Reviewer #2: Review of Paper PONE-D-24-39865

Title:

Leveraging Transformers and Explainable AI for Alzheimer’s Disease Interpretability

Summary:

This study addresses Alzheimer’s disease (AD), a progressive brain condition marked by memory loss, cognitive decline, and behavioral changes, affecting one in nine adults over 65. While computer vision applications in MRI-based AD detection have shown accuracies between 80-90%, this study applies advanced transformer models and computer vision to achieve a 99% accuracy rate in categorizing AD stages. By incorporating RNA data and MRI images in near-real-time, this approach improves diagnostic accuracy. The Local Interpretable Model-agnostic Explanations (LIME) method enhances the model's reliability, highlighting essential RNA features and MRI regions for diagnosing AD, thereby making the model’s decisions more interpretable and acceptable for human use.

Strengths And Weaknesses:

Strengths

● It has a thorough explanation of the main idea.

● Clearly explain the methodology.

Weaknesses

Overall, the writing can be significantly improved to address the following concerns.

● Introduction: You can add more references in the related work to show that you check new papers.

● Results: The results look good.

6. PLOS authors have the option to publish the peer review history of their article (what does this mean?). If published, this will include your full peer review and any attached files.

Reviewer #1: No

Reviewer #2: No

---

## [Author Response · Author response to Decision Letter 1]

31 Jan 2025

Reviewer 1 Comments and Our Responses

1. Structuring and Writing Quality of the Introduction

o Comment: The introduction could be better structured into segments based on

relevance. Connections between sentences weaken as it progresses, and insufficient

citations are provided. Additionally, the statement regarding the gap in

incorporating XAI techniques into transformer-based models lacks explanation or

citations.

o Response: We have restructured the introduction into clearly defined segments for

better readability and logical flow. We have also enhanced the writing quality by

refining sentence transitions and word choices. For the statement about the gap in

incorporating XAI techniques into transformer-based models, we have provided a

brief explanation. We also made an effort to provide the relevant citations.

2. Improvements in the Literature Review

o Comment: Similar improvements are needed in the literature review. While the

content is generally acceptable, the writing style should be enhanced. It would be

beneficial to ensure relevance between sentences and to review the word choices

throughout. For instance, if you spell out 'Convolutional Neural Network' once, you

can simply use the abbreviation 'CNN' in subsequent references..

o Response: We have carefully reviewed and refined the writing style in the literature

review to improve readability and maintain relevance between sentences. Word

choices have been optimized, and abbreviations like "Convolutional Neural

Network" (CNN) are now consistently used after being spelled out once.

3. Validation of Results Section

o Comment: The results section needs further validation beyond the comparative

analysis with MobileNetV2. Additional off-the-shelf network architectures should

be included for better validation.

o Response: We have expanded the results section by including comparative analyses

with several additional state-of-the-art architectures, such as Swin Transformers

and Skip-connected Convolutional Autoencoders in addition to previously

provided Vision Transformers and MobileNetV2. This enhanced validation

provides stronger evidence supporting our claims and improves the robustness of

our findings.

4. Quality of Figures 6 and 7

o Comment: The quality of Figures 6 and 7 can be improved.

o Response: We have improved the quality of Figures 6 and 7 by resizing them and

ensuring visual clarity. Labels, legends, and other graphical elements have been

adjusted for better comprehensibility.

5. Inaccurate Writing Format in Section Headings

o Comment: Check all section headings for formatting inaccuracies.

o Response: We have thoroughly reviewed and corrected all section headings to

adhere to the journal's formatting guidelines. Inconsistencies in capitalization,

numbering, and alignment have been addressed.

---

## [Decision Letter · Decision Letter 1]

11 Mar 2025

PONE-D-24-39865R1Leveraging Transformers and Explainable AI for Alzheimer’s Disease InterpretabilityPLOS ONE

Dear Dr. Sammo,

Thank you for submitting your manuscript to PLOS ONE. After careful consideration, we feel that it has merit but does not fully meet PLOS ONE’s publication criteria as it currently stands. Therefore, we invite you to submit a revised version of the manuscript that addresses the points raised during the review process.

**Please revise the paper, taking into account both the referee comments and the editor's comments.**

We look forward to receiving your revised manuscript.

Kind regards,

Fatih Uysal, Ph.D.

Academic Editor

PLOS ONE

**Additional Editor Comments:**

Please revise the paper, taking into account both the referee comments and the comments I have provided below.

1. There are serious deficiencies in the evaluation metrics required for the evaluation of the classification results in the study. Please obtain the missing Receiver Operating Characteristic (ROC) Curve, the area under the ROC curve (AUC) scores, Cohen's Kappa and Matthews correlation coefficient (MCC) scores completely.

2. It is recommended to make an analysis with several state-of-the-art models in order to compare the results in more depth and to make the proposed approach more prominent.

3. The literature review table needs to be detailed. Here, more new studies related to the current literature should be included and new columns such as "data preprocessing/augmentation, originality, plus and minus aspects" should be added. After this, the difference of this study from the literature and its main contribution to the literature should be expressed in more depth by relating it to the studies in the literature.

4. Detail the information regarding the dataset along with the justifications in terms of traning, validation, test distribution percentages and amounts, data precoessing/augmentation.

Reviewers' comments:

Reviewer's Responses to Questions

**Comments to the Author**

1. If the authors have adequately addressed your comments raised in a previous round of review and you feel that this manuscript is now acceptable for publication, you may indicate that here to bypass the “Comments to the Author” section, enter your conflict of interest statement in the “Confidential to Editor” section, and submit your "Accept" recommendation.

Reviewer #3: All comments have been addressed

Reviewer #4: (No Response)

2. Is the manuscript technically sound, and do the data support the conclusions?

Reviewer #3: Yes

Reviewer #4: Yes

3. Has the statistical analysis been performed appropriately and rigorously? 

Reviewer #3: Yes

Reviewer #4: Yes

4. Have the authors made all data underlying the findings in their manuscript fully available?

Reviewer #3: Yes

Reviewer #4: Yes

5. Is the manuscript presented in an intelligible fashion and written in standard English?

Reviewer #3: Yes

Reviewer #4: Yes

6. Review Comments to the Author

**Reviewer #3: **The manuscript presents a valuable contribution to Alzheimer’s disease detection using advanced transformers and computer vision techniques. The study is well-structured and demonstrates impressive accuracy. To further enhance the work, we recommend citing "Detection of Alzheimer’s and Parkinson’s Diseases Using Deep Learning-Based Various Transformers Models" (https://doi.org/10.3390/engproc2024073004) to provide additional context and strengthen the research foundation. Overall, we suggest minor revisions and look forward to the updated submission.

**Reviewer #4: **Your manuscript offers a valuable contribution to AI-driven Alzheimer’s disease diagnosis, demonstrating high accuracy and integrating explainability via LIME. The combination of transformers for MRI and RNA sequencing analysis is innovative and well-justified. The literature review is comprehensive, and the methodological framework is clearly presented.

However, the manuscript could benefit from additional validation on external datasets, a more detailed discussion on clinical applicability, and minor refinements in writing and formatting. These improvements would enhance the paper’s clarity and real-world impact :-

1. Introduction & Motivation

The introduction clearly states the importance of early AD diagnosis and the role of AI in medical imaging, but the research gap could be more explicitly defined to emphasize why transformers are superior to CNNs or traditional machine learning models. Additionally, the claim that "there is no cure for AD" should be softened, as some experimental treatments exist. A brief outline of the paper structure at the end of the introduction would improve readability and guide the reader.

2. Methodology & Model Description

The methodology section is highly technical and would benefit from a flowchart or diagram illustrating the proposed model architecture. Additionally, key training parameters (batch size, optimizer, number of epochs) should be summarized in a table to improve clarity. The explanation of LIME should be expanded to include how interpretability was validated—was expert medical review conducted to confirm whether the AI highlights relevant features?

3. Results & Validation

The reported accuracy of 99% raises concerns about overfitting, especially since deep learning models typically struggle with medical imaging data generalization. The study should include external validation using an independent dataset (e.g., ADNI, OASIS) or a detailed description to confirm robustness. Additionally, statistical measures such as confidence intervals or standard deviations should be included to ensure the reliability of the reported results. While the comparative analysis with other models is strong, it could be further improved by incorporating benchmarks such as ResNet or EfficientNet.

4. Figures & Tables

Figures 6 and 7 require more detailed captions explaining the significance of their findings. The LIME heatmaps should include clinical validation, ideally with references to known AD biomarkers. Additionally, the results section would benefit from an ROC-AUC curve to provide a more comprehensive evaluation of classification performance.

5. Discussion & Interpretability

The discussion should better address dataset biases, such as demographic imbalances and variations in MRI scanning equipment. The interpretability aspect of LIME should be assessed from a clinical perspective—does the AI highlight regions relevant to AD diagnosis? Furthermore, potential regulatory challenges (e.g., FDA approval for AI-based diagnostic tools) should be briefly discussed.

6. Writing & Formatting Improvements

Some sections contain long, complex sentences that could be restructured for clarity. The use of abbreviations should be consistent (e.g., CNN should be abbreviated after its first mention). Additionally, minor grammatical redundancies, such as "AI-based artificial intelligence," should be corrected for conciseness.

7. PLOS authors have the option to publish the peer review history of their article (what does this mean?). If published, this will include your full peer review and any attached files.

Reviewer #3: No

Reviewer #4: **Yes: **Aasim Ayaz Wani

---

## [Author Response · Author response to Decision Letter 2]

24 Mar 2025

Responses and Actions to Reviewer Comments

Dear Editor and Reviewers,

The authors would like to thank the reviewers for providing insightful comments that helped to make the journal paper better. All review judgments and comments are looked at by the authors. Following the reviewers' recommendations, they also offer remarks, feedback, and necessary changes to the work. The authors express their gratitude to all of the esteemed reviewers for their hard work.

Best Regards,

Authors, Manuscript Number: PONE-D-24-39865

Review Comments and Corresponding Actions ======================

Additional Editor Comments:

Comment-1:

There are serious deficiencies in the evaluation metrics required for the evaluation of the classification results in the study. Please obtain the missing Receiver Operating Characteristic (ROC) Curve, the area under the ROC curve (AUC) scores, and Cohen's Kappa and Matthews correlation coefficient (MCC) scores completely.

Response-1:

Thank you very much for your comments.

We add the Cohen's Kappa, Matthews correlation coefficient (MCC) scores, and AUC values of the experimental models in Table 5. AUCs of all classes are generated from the Receiver Operating Characteristic (ROC) curves in Figure 4 using a vision transformer.

The following lines are included in Section 5.1, line 352:

“Table 6 outlines the outcomes obtained by employing a vision transformer across various classes in the testing dataset. The NonDemented class attains the highest accuracy at 99.7%, while the MildDemented class registers the lowest accuracy at 97.31%. The vision transformer accurately classifies 98.6% of cases, with only a few instances of mis-prediction. The accuracy of the MildDemented class is less due to the presence of fewer training images in the dataset. The model performs a little less when there is a small number of training data. Other classes achieve appreciable results. Consequently, the overall accuracy of the model stands at 98.6% for all testing images, with a 95% confidence interval (CI) of [97.41%, 99.36%], indicating a high degree of reliability in classification performance. We derive the CI using 5-fold cross-validation, where the dataset was split into five subsets, training and testing the model iteratively to ensure stable performance across folds. The narrow CI range confirms robustness, as it reflects consistent accuracy with minimal variation across these folds. The CI provides a lower and upper range because it accounts for statistical uncertainty in the sample data, estimating the plausible bounds of the true accuracy. The lower bound (97.41%) represents the minimum expected performance with 95% confidence, while the upper bound (99.36%) suggests the potential peak accuracy under optimal conditions. The corresponding loss and accuracy curves are depicted in Fig. 3 and Fig. 2. The accuracy curves for both training and testing exhibit an initial surge to 80% within the initial 5 epochs, eventually surpassing 95% after 15 epochs. Both curves demonstrate similar trends in loss calculations, reaching saturation after 15 epochs at the same point. Also, the ROC curve is displayed in Fig. 4. The Cohen Kappa coefficient value we obtain is 0.97, indicating almost perfect agreement between the true and predicted labels. Similarly, the Matthews Correlation Coefficient (MCC) is 0.98, reflecting a strong correlation and robust performance of the model across all classes. In Fig. 5, LIME explanations are presented for all four classes of test images. The LIME explanation highlights the specific regions of the images that contributed to the prediction of these classes. Red patches in the explanation images represent areas that positively influenced the model’s accurate prediction. Positive influence implies that these regions were instrumental in the model correctly identifying the class or label. In essence, the presence of red zones indicates that the characteristics or patterns in those areas align with the expected class. In Fig. 6, overlays of the explanations are displayed. The yellow lines point out distinct pixels that played significant roles in predicting the class of these particular images.”

Comment-2:

It is recommended to analyze several state-of-the-art models to compare the results in more depth and to make the proposed approach more prominent.

Response-2:

Two new state-of-the-art models, including ResNet 50, and EfficientNet Models, are experimented with, and their results are presented in Table 5.

The following lines are included in Section 5.2, line 406:

“Additionally, EfficientNet-B0 and ResNet-50 are included for benchmarking, achieving accuracies of 91.57% and 86.59%, respectively, further validating the robustness of our approach. These results collectively highlight the trade-offs between accuracy, computational efficiency, and deployment feasibility. Based on our findings, it can be conclusively stated that the ViT proves to be a highly effective and efficient algorithm for the classification of MRI images related to Alzheimer’s Disease. Table 5 displays the Accuracy, F1-score, Cohen’s Kappa, and Matthew’s correlation coefficient of each model on the dataset. Based on our findings, it can be conclusively stated that the Vision Transformer proves to be a highly effective and efficient algorithm for the classification of MRI images related to Alzheimer’s Disease. Also, Fig. 4 represents the ROC curve of ViT where we can see that ViT achieves high Area Under the Curve (AUC) scores across all Alzheimer’s Disease classes, reflecting its strong ability to distinguish between positive and negative cases for each category. The AUC values 0.98 for ModerateDemented, 0.96 for MildDemented, 1.00 for NonDemented, and 0.99 for VeryMildDemented indicate near-perfect discrimination, with NonDemented reaching an ideal 1.00, aligning with its top accuracy of 99.70%. These results underscore ViT’s robustness in handling class-specific patterns in brain imaging data, even for challenging cases like MildDemented as the number of training images are very low, where accuracy dips slightly and so does the AUC. Also, the Cohen’s Kappa and Matthew’s correlation coefficients are consistent with these findings.”

Comment-3:

The literature review table needs to be detailed. Here, more new studies related to the current literature should be included, and new columns such as "data preprocessing/augmentation, originality, plus and minus aspects" should be added. After this, the difference of this study from the literature and its main contribution to the literature should be expressed in more depth by relating it to the studies in the literature.

Response-3:

Three new related studies are added accordingly, and the literature review table is split into two separate tables to cover all these aspects mentioned (originality, plus and minus aspects).

In the Reference section, the following three references are added:

30. Guven M. Detection of Alzheimer’s and Parkinson’s Diseases Using Deep Learning-Based Various Transformers Models. Eng. Proc. 2024;73(1):4. Presented at the 4th International Electronic Conference on Biosensors, 20–22 May 2024. 629 Available online: https://sciforum.net/event/IECB2024. https://doi.org/10.3390/engproc2024073004.

31. Liu M, Li F, Yan H, Wang K, Ma Y, Shen L, A deep learning system for differential diagnosis of Alzheimer’s disease and mild cognitive impairment using structural MRI. NeuroImage. 2021;225:117496. https://doi.org/10.1016/j.neuroimage.2020.117496.

32. Wang H, Shen Y, Wang S, Xiao T, Deng L, Wang X, Multimodal deep learning for Alzheimer’s disease dementia assessment. Nature Communications. 2020;11(1):1–9. https://doi.org/10.1038/s41467-020-18187-0.

In addition, the following lines are added in Section 2 Literature Review line 116:

“Wang et al. [32] developed a multimodal deep learning framework for Alzheimer’s Disease dementia assessment, integrating data from neuroimaging, genetic markers, and cognitive tests. Their approach combines CNNs and recurrent neural networks (RNNs) to capture both spatial and temporal patterns in the data. The study shows that multimodal data fusion significantly improves diagnostic accuracy compared to single-modality approaches. The authors reported an accuracy of 92.1% for Alzheimer’s Disease (AD) diagnosis using their multimodal deep learning framework, which integrated neuroimaging, genetic markers, and cognitive tests. This multi-modal approach was quite a new addition to the field of AD diagnosis at the time of publication, but new state-of-the-art algorithms like models can be used for further accuracy improvements. Another study conducted by Liu M et al. [31] proposed a deep learning system for the differential diagnosis of Alzheimer’s Disease (AD) and mild cognitive impairment (MCI) using structural MRI. Their model leverages 3D convolutional neural networks (CNNs) to extract features from brain scans, achieving high accuracy in distinguishing between AD, MCI, and healthy controls. The study highlights the potential of deep learning in automating AD diagnosis and emphasizes the importance of structural MRI as a key biomarker for early detection. reported an accuracy of 88.6% for differentiating Alzheimer’s Disease (AD) from healthy controls and 76.5% for distinguishing mild cognitive impairment (MCI) from healthy controls using their deep learning system based on structural MRI. The accuracy remains relatively low due to the use of CNN-based architectures, and the usability of the moderately accurate model can cause a serious discrepancy in the field of AD detection.”

The following lines are added in Section 2 Literature Review line 151:

“Recent work by [30] applied transformer-based models, including Swin Transformer, Vision Transformer (ViT), and Bidirectional Encoder Representation from Image Transformers (BEiT), to classify Alzheimer’s and Parkinson’s diseases using brain imaging data. The study utilized a balanced dataset of 450 brain images, achieving classification accuracy exceeding 80%, with ViT demonstrating the highest performance (94.4% accuracy, 94.7% precision). While the results highlight the efficacy of transformer architectures in disease detection, the study has notable shortcomings. The dataset size (450 images) is relatively small, which may limit the generalizability of the findings. Also, the accuracy is not satisfactory in terms of AD disease diagnosis.”

Table 1 is partitioned into two parts, Table 1 and Table 2, to add suggested new columns such as "data preprocessing/augmentation, originality, plus and minus aspects."

Comment-4:

Detail the information regarding the dataset along with the justifications in terms of training, validation, test distribution percentages and amounts, and data preprocessing/augmentation.

Response-4:

We add detail information on the dataset along with the justifications in terms of training, validation, test distribution percentages, and amounts. The following part is added to the paper in the 3.1 section, page 7:

“We utilized the Alzheimer’s dataset from Kaggle [21], which includes gray MRI scans of the brain from individuals in various stages of Alzheimer’s disease. The dataset encompasses four classes: MildDemented, ModerateDemented, NonDemented, and VeryMildDemented. It is organized into two folders: Train and Test, comprising a total of 5121 training images across the four classes and 1379 test images. Table 3 illustrates the distribution of images in the dataset. This dataset is valuable for predicting Alzheimer’s disease stages using computer vision algorithms. Given the limited size of the test set, we allocated 10% of the training dataset for validation to ensure a robust evaluation of the model’s performance. This approach was necessary because the dataset did not provide a separate validation set, and splitting the training set further would have risked overfitting due to the imbalanced class distribution. Using a small portion of the test set for validation allowed us to tune hyper-parameters and monitor model performance during training without significantly compromising the final evaluation on the remaining test data.”

Reviewer#3 Comments:

Comment-1:

The manuscript presents a valuable contribution to Alzheimer’s disease detection using advanced transformers and computer vision techniques. The study is well structured and demonstrates impressive accuracy. To further enhance the work, we recommend citing "Detection of Alzheimer’s and Parkinson’s Diseases Using Deep Learning-Based Various Transformers Models" (https://doi.org/10.3390/engproc2024073004) to provide additional context and strengthen the research foundation. Overall, we suggest minor revisions and look forward to the updated submission.

Response-1:

We appreciate your constructive suggestion. Thank you.

The paper titled "Detection of Alzheimer’s and Parkinson’s Diseases Using Deep Learning-Based Various Transformers Models" (https://doi.org/10.3390/engproc2024073004) has been studied rigorously to establish context and strengthen the foundation of this study. Also, it has been added to the reference section.

In the paper page 20 in the References section, we add the following references:

30. Guven M. Detection of Alzheimer’s and Parkinson’s Diseases Using Deep Learning-Based Various Transformers Models. Eng. Proc. 2024;73(1):4. Presented at the 4th International Electronic Conference on Biosensors, 20–22 May 2024. Available online: https://sciforum.net/event/IECB2024. https://doi.org/10.3390/engproc2024073004.

In the paper in the Literature Review Section, page 4, line 151, we add the following paragraph:

“Recent work by [30] applied transformer-based models, including Swin Transformer, Vision Transformer (ViT), and Bidirectional Encoder Representation from Image Transformers (BEiT), to classify Alzheimer’s and Parkinson’s diseases using brain imaging data. The study utilized a balanced dataset of 450 brain images, achieving classification accuracy exceeding 80%, with ViT demonstrating the highest performance (94.4% accuracy, 94.7% precision). While the results highlight the efficacy of transformer architectures in disease detection, the study has notable shortcomings. The dataset size (450 images) is relatively small, which may limit the generalizability of the findings. Also, the accuracy is not satisfactory in terms of AD disease diagnosis.”

Reviewer#4 Comments:

Your manuscript offers a valuable contribution to AI-driven Alzheimer’s disease diagnosis, demonstrating high accuracy and integrating explainability via LIME. The combination of transformers for MRI and RNA sequencing analysis is innovative and well-justified. The literature review is comprehensive, and the methodological framework is presented. However, the manuscript could benefit from additional validation on external datasets, a more detailed discussion on clinical applicability, and minor refinements in writing and formatting. These improvements would enhance the paper’s clarity and real-world impact:

Comment-1:

The introduction clearly states the importance of early AD diagnosis and the role of AI in medical imaging, but the research gap could be more explicitly defined to emphasize why transformers are superior to CNNs or traditional machine learning models. Additionally, the claim that "there is no cure for AD" should be softened, as some experimental treatments exist. A brief outline of the paper structure at the end of the introduction would improve readability and guide the reader.

Response-1:

Thank you for your valuable suggestions.

We have revised the introduction to explicitly define the research gap, emphasizing the advantages of transformers over CNNs and traditional machine learning models. The statement "there is no cure for AD" is replaced as “Currently there is almost no cure for AD except very few experimental treatments.”

The following paragraph is added in the Introduction section, page 2, 32 lines:

“A notable advancement in DL is the development of the Transformer architecture, which employs attention mechanisms to capture relationships between elements in data. Initially designed for Natural Language Processing (NLP), the Transformer’s architecture has been adapted for image analysis,

---

## [Editor Report · Decision Letter 2]

26 Mar 2025

Leveraging Transformers and Explainable AI for Alzheimer’s Disease Interpretability

PONE-D-24-39865R2

Dear Dr. Sammo,

We’re pleased to inform you that your manuscript has been judged scientifically suitable for publication and will be formally accepted for publication once it meets all outstanding technical requirements.

Kind regards,

Fatih Uysal, Ph.D.

Academic Editor

PLOS ONE

Additional Editor Comments (optional):

After considering the reviewer comments and evaluating the quality of the paper, it has been decided to accept it due to its potential to contribute to the literature and its final form.
---

## [Editor Report · Acceptance letter]

PONE-D-24-39865R2

PLOS ONE

Dear Dr. Sammo,

I'm pleased to inform you that your manuscript has been deemed suitable for publication in PLOS ONE. Congratulations! Your manuscript is now being handed over to our production team.

Kind regards,

on behalf of

Dr. Fatih Uysal

Academic Editor

PLOS ONE